# The Biological Function of Genome Organization

**DOI:** 10.3390/ijms26189058

**Published:** 2025-09-17

**Authors:** Xin Yang, Hongni Zhu, Yajie Liu, Jinhong Wang, Yi Song, Shasha Liao, Peng Dong

**Affiliations:** 1Institute of Biomedical and Health Engineering, Shenzhen Institutes of Advanced Technology, Chinese Academy of Sciences, Shenzhen 518055, China; x.yang2@siat.ac.cn (X.Y.); hn.zhu@siat.ac.cn (H.Z.); yj.liu5@siat.ac.cn (Y.L.); jh.wang2@siat.ac.cn (J.W.); yi.song@siat.ac.cn (Y.S.); ss.liao@siat.ac.cn (S.L.); 2State Key Laboratory of Biomedical Imaging Science and System, Shenzhen Institutes of Advanced Technology, Chinese Academy of Sciences, Shenzhen 518055, China

**Keywords:** genome architecture, genome organization, chromatin loop, topologically associating domain, compartment

## Abstract

The mammalian genome is hierarchically packaged into distinct functional units, including chromatin loops, topologically associating domains, compartments and chromosome territories. This structural organization is fundamentally important because it orchestrates essential nuclear functions that underpin normal cellular identity and organismal development. In this review, we synthesize current understanding of the intricate relationship between genome architecture and its critical biological roles. We discuss how hierarchical structures are dynamically established and maintained by architectural proteins, transcription factors, epigenetic regulators and non-coding RNAs via distinct mechanisms. Importantly, we focus on the functional consequences of three-dimensional (3D) genome organization and discuss how it modulates fundamental biological processes such as transcription, gene co-expression, epigenetic modification, DNA replication and repair. We also examine the dynamics of 3D genome organization during cellular differentiation, early embryonic development and organogenesis, followed by discussing how structural disruptions are mechanistically linked to various human diseases. Understanding the biological function of 3D genome organization is thus not only essential for deciphering fundamental nuclear processes but also holds significant promise for elucidating disease etiologies and developing effective therapeutics.

## 1. Introduction

In human cells, the total length of genomic DNA approaches two meters, whereas the nuclear diameter is only a few micrometers. To compact this vast amount of genetic material into a confined nuclear space while preserving dynamic access and regulation of genetic information, the genome is intricately folded and organized into a highly ordered three-dimensional (3D, see Appendix A for major acronyms and their corresponding full names) architecture, comprising chromosome territories, nuclear compartments, topologically associating domains (TADs), and chromatin loops [1,2]. This spatial organization is orchestrated by the coordinated action of chromatin structural proteins, such as CTCF and cohesin [3,4,5,6], together with various transcriptional and epigenetic mechanisms [7,8]. Such 3D folding not only overcomes the physical constraints of DNA packaging but also establishes the fundamental framework for genomic function, ensuring precise regulation of gene expression across different developmental stages and environmental conditions [1,2,9].

For a long period of time, due to technical limitations, our understanding of the 3D genome architecture remained rudimentary. Chromatin was traditionally simplified as a linear sequence or modeled as a randomly coiled polymer, overlooking its spatial complexity. The development of high-throughput chromatin conformation capture (3C) techniques such as Hi-C [10], Micro-C [11], and CHIA-PET [12] have greatly expanded the study of genomics in 3D space. Specifically, Hi-C and Micro-C have enabled the delineation of several spatial organizational features, including chromatin loops [13], TADs [14] and chromosomal compartments [10]. In contrast, CHIA-PET integrates chromatin interaction data with protein–DNA binding information, allowing researchers to focus on specific regulatory interactions mediated by transcription factors or architectural proteins, such as enhancer–promoter (E-P) loops. Although 3C techniques can provide sequence-specific mutual contact information, they are indirect and usually involved pooled measurements, limiting their ability of mapping global spatial distribution. To address these issues, a number of approaches based on single-molecule localization imaging were developed for either tracing defined chromosomal regions or mapping global distribution of *cis*-regulatory elements [15,16,17]. These imaging-based approaches have enabled researchers to directly visualize chromatin folding at nanometer-scale resolution, facilitating investigations of chromatin positioning, compaction, and long-range interactions at the single-cell level [17,18,19]. To date, the integration of analyses based on both sequencing and imaging approaches has revealed most of the details about genome folding, highlighting the fact that the eukaryotic genome is not randomly wrapped within the nucleus but hierarchically folded into multiple structural layers.

The precise 3D organization of the genome is fundamentally important because it orchestrates essential regulations that maintain cellular function and identity [20,21]. Far beyond simple packaging, this intricate spatial architecture directly governs critical processes by strategically bringing regulatory elements together or keeping them apart. It is indispensable for the precise spatiotemporal control of gene expression during development and differentiation, enabling specific enhancers to activate their target promoters while insulating inappropriate interactions [22,23]. Furthermore, 3D organization facilitates the coordinated timing of DNA replication, enhances the efficiency and accuracy of DNA repair, maintains chromosome stability by preventing harmful translocations, and provides a structural framework for the establishment and maintenance of cell-type-specific epigenetic programs [8,24,25,26,27]. However, dysregulation of this architectural framework, driven by mutations in structural proteins like CTCF or cohesin, frequently leads to aberrant gene expression and genomic instability, directly contributing to developmental disorders, cancer, and other human diseases [21,28,29]. 

In this review, we first discuss recent discoveries regarding the principles of spatial genome organization and the molecular and physical mechanisms underlying its regulation. We then focus on the important cellular function of 3D genome structure in regulating gene transcription, in particular coordinating enhancer–promoter interaction and gene co-expression. We further examine the relationship between genome structure and other molecular events, including epigenetic modification, DNA replication and DNA damage repair. Finally, we discuss the latest findings on the role of genome folding during cellular differentiation and developmental processes, and its alterations related to various human diseases. In conclusion, we highlight the key questions that remain to be addressed in this rapidly developing field.

## 2. Hierarchical Genome Folding and Compaction

The hierarchical folding and compaction of the genome are essential for accommodating the vast amount of DNA within the limited space of the cell nucleus. At the same time, this structural organization supports critical biological processes such as transcription, replication, and repair. These mechanisms are vital for the regulation of gene expression, preservation of genome stability, and execution of various cellular functions.

### 2.1. Chromatin Loops

Among hierarchical genome structures, chromatin loops are minimal structural components (Figure 1). Ring-shaped cohesin complexes are loaded onto chromatin by loader proteins and act as molecular motors to extrude DNA loops through their ring structure by utilizing energy from ATP hydrolysis [30]. This extrusion is stabilized by CTCF protein, the binding of which to DNA serves as an anchor point for the extrusion complex [6]. These chromatin loops are mostly E-P interactions and appear as “peaks” of high contact frequency in Hi-C contact matrices [31]. Apart from cohesin-mediated short-range chromatin loops, Hi-C experiments also detect long-range chromatin interactions that bring together sequences separated by hundreds of kilobases or even megabases along the linear chromosome, enabling critical regulatory functions beyond simple linear proximity [32,33,34]. Interestingly, a systematic survey of long-range chromatin interactions centered on 18,943 well-annotated promoters for protein-coding genes across 27 human cell or tissue types revealed that, a considerable fraction of these interactions occurs between two promoters. This observation strongly suggests that distinct mechanisms underlie the formation of long-range chromatin loops [35].

Recent studies have emphasized the nature of chromatin loops as both dynamic and conserved. A novel integrative method combining Micro-C and live-cell imaging has provided absolute genome-wide quantification of chromatin loops, demonstrating their low formation probability and dramatic heterogeneity across different interaction types [36,37]. Furthermore, Micro-C analyses have revealed the presence of chromatin loops in the cnidarian *Nematostella vectensis*, with certain loops spanning nearly one megabase, suggesting that chromatin looping is a conserved feature across metazoan lineages [38].

### 2.2. Topologically Associating Domains

Topologically associating domains (TADs), which represent large-scale structural units encompassing multiple chromatin loops, are genomic regions characterized by a higher frequency of intra-domain chromatin interactions than that of inter-domain interactions (Figure 1) [39]. TADs serve as stable organizational units of the chromosome and primarily act as independent regulatory landscapes that ensure the proper spatial segregation of gene regulatory activities [40]. Within TADs, chromatin regions display strong internal interactions, whereas boundaries enriched with cohesin and CTCF proteins act as insulators, preventing aberrant E-P interactions and maintaining appropriate gene regulation. Recently, it was shown that deletion of an ultra-conserved TAD boundary in mice resulted in altered gene expression and consequentially subtle yet significant developmental defect in cardiac tissue [41], underscoring the functional importance of TAD boundaries in forming such insulations. Interestingly, TAD boundaries are evolutionarily conserved, since comparative studies across eight mammalian species using Hi-C and ChIP-seq data have identified ultra-conserved TAD boundaries characterized by stronger insulation and higher CTCF occupancy [42]. However, species-specific boundaries are enriched for active transcription and bivalent chromatin marks, suggesting their primary role in coordinating cell differentiation and lineage specification.

### 2.3. Compartments

At a broader level of chromatin organization, TADs are nested within larger structural units known as compartments. Each chromosome is partitioned into two types of compartments—A and B (Figure 1). The A compartment is associated with actively transcribed genes and is characterized by open chromatin conformation, high gene density, and enrichment of transcriptional regulatory elements [43,44,45]. In contrast, the B compartment contains transcriptionally repressed genes and is marked by compact chromatin structure and specific histone modifications that restrict access to transcription factors and RNA polymerase [43,45]. Compartments reflect the spatial arrangement of the genome and are closely linked to chromatin states and gene expression patterns. It is notable that the A/B compartmentalization is dynamic and undergoes reorganization in response to developmental cues and environmental signals. For example, the study on the genome architecture of *Crassostrea ariakensis* has revealed substantial compartmental shifts following environmental transplantation, underscoring the influence of external stimuli on chromatin compartmentalization [45].

### 2.4. Chromosome Territories

Lastly, individual interphase chromosomes preferentially occupy distinct, spatially segregated territories within the nucleus (Figure 1) [46]. This organizational principle is well-established through microscopy-based techniques, such as chromosome painting and 3C-based sequencing technologies [47,48]. However, this segregation is not absolute, as different chromosomes frequently intermingle with each other. Notably, within each chromosome territory, the spatial positioning of genomic loci is non-random and closely correlates with transcriptional activity [49]. Moreover, gene-rich regions poised for transcription frequently localize near the periphery of chromosome territories, suggesting a positional effect on transcriptional activation [49].

## 3. Regulation of 3D Genome Architecture

Genome folding is a highly complex and tightly regulated process governed by a variety of regulators, including chromatin architectural proteins, transcription factors, epigenetic regulations and even non-coding RNAs. These regulators collectively shape the 3D architecture of the genome through diverse biochemical and biophysical mechanisms.

### 3.1. Architectural Proteins and the Loop Extrusion Mechanism

Nature has evolved a set of architectural proteins that orchestrate genome folding, among which the most extensively studied are cohesin complex and CTCF. Cohesin was initially identified as a critical regulator of sister chromatid cohesion during mitosis [50], whereas CTCF was originally characterized as an insulator protein, capable of restricting E-P interactions that modulate gene transcription [51,52]. During interphase, cohesin interacts with CTCF and other factors and is proposed to regulate genome folding through a loop extrusion mechanism (Figure 2A). The loop extrusion model posits that structural maintenance of chromosome (SMC) protein complexes, in particular cohesin complex, extrudes DNA loops by progressively translocating along the chromatin fiber [3,4,53]. Mechanistically, cohesin mediates the juxtaposition of chromatin segments, while CTCF localizes cohesin to loop anchor sites and stabilizes loop structures by preventing cohesin from premature dissociation [54,55], thereby contributing to the establishment of TAD boundaries [6,56]. In consistent with this model, functional studies have shown that CTCF depletion increases inter-TAD chromatin loops at the expense of intra-TAD loops [57], whereas depletion of cohesin subunits (e.g., RAD21) significantly reduces chromatin looping both within and between TADs [58,59]. Notably, additional regulators such as the bromodomain and extra terminal domain (BET) family protein BRD2 can also serve as architectural proteins [60]. BRD2 promotes both spatial mixing and compartmentalization of active chromatin. This activity does not rely on transcription but depends on BRD2’s double bromodomain for acetylated target recognition, as well as its intrinsic disordered region for protein interactions [60].

Recently, through directly visualizing chromatin looping at the *Fbn2* TAD in mouse embryonic stem cells (ESCs), Gabriele et al. quantified looping dynamics through Bayesian inference and found that the well-defined chromatin loops are unexpectedly rare and dynamic [37]. Based on this direct measurement, they proposed a “dynamic” loop maintenance complex (LMC) model, in which CTCF recruits cohesin to anchor sites, where cohesin holds chromatin loops together while molecules dynamically bind and unbind. These findings indicate that cohesin-mediated loops are transient, undergoing continuous formation and disassembly throughout the cell cycle, with average lifetimes ranging from several to tens of minutes [61].

### 3.2. Transcription Factors and Phase Separation Mechanism

Transcription factors (TFs) and co-factors possess DNA-binding capacity but generally exert their regulatory functions via interactions with other proteins. These interactions have the potential to affect genome conformation. While TFs can enhance the formation of DNA loops via direct self-association or by recruiting cofactors that in turn form oligomers [62,63], increasing evidence suggests that most of them assemble condensates through weak but multivalent interactions, exhibiting characteristics of liquid–liquid phase separation (Figure 2B). During cell fate transitions, transcription factors such as OCT4 undergo phase separation to form biomolecular condensates, which subsequentially reorganize chromatin loops and TAD structures [64]. Disruption of OCT4 condensates impairs TAD reorganization and hinders cellular reprogramming. In another example, analyses by using single-molecule Förster resonance energy transfer (FRET) and fluorescence cross-correlation spectroscopy (FCCS) demonstrated that oligomerization of human NANOG protein is essential for bridging DNA elements in vitro [65]. The study further confirmed that assembly of NANOG prion-like domain is fundamental for specific DNA recognition and distant chromatin interactions. On the other hand, the default transcription coactivator, Mediator, has been proposed to interact with cohesin in order to bring enhancers and promoters into physical proximity. Supporting this notion, depletion of Mediator significantly reduces E-P interaction frequencies, accompanied by considerable decrease in gene expression [66,67]. Further studies employing super-resolution imaging and light-activated optoDroplet assay have revealed that Mediator proteins can form liquid-like condensates at super-enhancers (SEs), thereby compartmentalizing and concentrating the transcription apparatus from nuclear extracts [68,69]. These findings support a model in which coactivator condensates at SEs, driven by their intrinsically disordered regions, spatially enrich the transcription components to ensure the robust activation of cell-identity genes.

### 3.3. Transcriptional Activity

Transcription has long been considered a critical determinant of shaping genome architecture, owing to its pervasive activity across the genome and its capacity to modulate the local chromatin landscape [70]. Firstly, compartmentalization into A and B chromatin regions represents the most prominent level of genome organization linked to transcriptional activity. While the formation of inactive B compartment depends on HP1 protein and L1 repeat RNA, active A compartments self-assemble via multivalent interactions involving RNA polymerase II (Pol II), transcription factors and co-factors, splicing factors, nascent RNA-binding proteins, and histone modifications [71,72,73]. Beyond compartmentalization, transcription can influence chromatin topology by acting as a moving barrier to loop extrusion. Although TAD boundaries are typically defined by CTCF binding sites, highly transcribed genes are also enriched at insulating boundaries and can serve as alternative barriers [39,74]. Notably, these transcription-associated structures are diminished upon transcriptional inhibition or cohesion depletion, but become more pronounced in the absence of CTCF, suggesting a direct interplay between transcriptional activities and chromatin architectural proteins in shaping genome topology.

### 3.4. Epigenetic Regulations

Epigenetic regulations can shape genome structure by modulating chromatin accessibility and recruiting architectural proteins such as CTCF, HP1, and Polycomb repressive complexes (PRCs) [75,76,77]. Specific histone marks actively modulate chromatin folding by orchestrating the localization and activity of chromatin remodelers. For example, repressive histone mark H3K27me3 facilitates the recruitment of PRC1, which suppresses chromatin loop formation by establishing repressive chromatin domains and promoting local compaction [78,79]. This mark also enables PRC1 to undergo phase separation, forming biomolecular condensates that concentrate its E3 ubiquitin ligase subunit RNF2 and facilitate the deposition of H2AK119ub1, thereby reinforcing transcriptional silencing and heterochromatin compaction. Similarly, H3K9me3 mediates the recruitment of HP1 proteins to pericentromeric regions, where they dimerize and coalesce into liquid-like condensates [80,81]. These condensates are estimated to be 100~1000 nm in diameter, which can mediate long-range interactions by bringing distal heterochromatic regions into proximity. In contrast, active histone acetylation marks, catalyzed by histone acetyltransferases (HATs), which can neutralize positive charges on lysine residues. This electrostatic alteration weakens DNA-histone interactions, thereby facilitating chromatin relaxation and enhancing accessibility [82]. For instance, H4K16ac prevents the formation of condensed 30-nm fibers and higher-order chromatin folding [83], and H3K9ac is often associated with transcriptionally active regions [84]. Conversely, histone deacetylases such as SIRT1/2 promote DNA compaction by removing H4K16 acetylation and reinforce heterochromatin formation through H3K9me3 deposition and HP1 recruitment [85,86], potentially functioning as a global regulator of 3D genome structures, in particular compartments. Recent advances in super-resolution imaging have provided deeper insight into the spatial distribution of epigenetic states at nanometer scales. In *Drosophila spermatocytes*, histone marks such as H3K4me3, H3K27me3, and H3K36me3 have been shown to form spatially discrete and mutually exclusive nucleosome clusters along chromatin loops [87]. These modifications correlate with distinct transcriptional activities, indicating that histone marks serve not only as biochemical signals but also as structural cues that help organize chromatin into spatially and functionally specialized domains.

On the other hand, DNA methylation, which is predominantly localized at CpG dinucleotides, modulates 3D genome architecture by impeding transcription factor binding and promoting chromatin condensation through the recruitment of methyl-CpG-binding proteins [88,89]. This methylation enhances intra-chromosomal interactions, reduces inter-chromosomal contacts and induces centromeric compaction, thereby limiting overall chromosomal flexibility [88]. Taken together, these diverse mechanisms pinpoint the important role of epigenetic regulations in maintaining chromatin organization and, consequently, in precisely controlling gene expression and cellular identity.

### 3.5. Non-Coding RNAs

Long non-coding RNAs (lncRNAs) play crucial roles in shaping chromatin conformation. A well-studied example is Xist lncRNA, a *cis*-acting architectural RNA that rewires the 3D structure of the entire X chromosome [90]. Xist functions by recruiting repressive complexes, disrupting active chromatin domains, and facilitating the formation of large condensed compartments, potentially through phase separation mechanism. These coordinated actions convert the active X chromosome into a tightly folded and transcriptionally inert Barr body, thereby ensuring precise dosage compensation in female mammals [91,92]. Recent findings further reveal that Xist can nucleate supramolecular complexes (SMACs), which consist of many copies of the key silencing protein SPEN [92]. Aggregation and exchange of SMAC proteins generate local protein gradients, which in turn drive chromatin compaction and stable gene silencing.

## 4. Biological Function of 3D Genome Architecture

### 4.1. Enhancer-Promoter Interaction and Gene Transcription

Enhancers are DNA regulatory elements that can significantly boost gene transcription, regardless of their orientation or linear distance from the promoter [67]. These elements can be resided upstream, downstream, or within introns of their target genes, often exerting their regulatory effects on promoters across distances ranging from several kilobases to hundreds of kilobases [93,94]. A key mechanism underlying such long-range E-P interactions is chromatin looping, which is largely facilitated by architectural proteins such as cohesin and CTCF [5,35,93,94,95]. In addition to structural looping, transcription factors and co-factors also play essential roles in stabilizing E-P interactions [93]. For example, proteins such as YY1 can simultaneously bind to both enhancer and promoter sequences, thereby serving as a molecular bridge to stabilize their interaction [62]. Moreover, RNA-binding proteins can facilitate E-P communication by binding nascent transcripts at both sites and mediating RNA-RNA or RNA-protein interactions that tether enhancers to their promoters [96]. Interestingly, by generating high-confidence E-P RNA interaction maps using RNA in situ conformation sequencing technology, Liang et al. discovered that 37.9% of the E-P RNA interaction sites overlap with short interspersed nuclear elements, particularly Alu sequences. This finding suggests specific genomic sequences may serve as determinant of E-P interaction selectivity within the genome [97].

The regulation of E-P interactions by 3D genome organization plays a critical role in directing how disease-risk single nucleotide variants (SNVs) influence distal molecular phenotypes. These SNVs frequently reside in distal *cis*-regulatory elements such as enhancers, and can modulate complex traits or disease susceptibility by altering chromatin states at interacting regulatory elements. For instance, a recent HiChIP study revealed that 684 autoimmune disease-associated intergenic SNVs can influence the expression of 2597 target genes through E-P interactions, with a single SNV capable of affecting up to ten genes [98,99]. Disruption of 3D genome architecture caused by SNVs contribute to disease phenotypes primarily via two mechanisms. In the first mechanism, disease-associated regulatory SNVs disrupt physiological E-P interactions or establish pathogenic ones, resulting in aberrant gene expression and disease manifestations. A case in point is the 3q rearrangements, which repositions a distal *GATA2* enhancer to ectopically activate EVI1, and simultaneously confer GATA2 functional haploinsufficiency, thereby contributing to sporadic occurrence of familial acute myeloid leukemia [100]. Similarly, duplication at *Xq26.1* locus alters the interactions between flanking genes and putative enhancers, leading to the overexpression of ARHGAP36 in hair follicles, basal cell carcinoma and trichoepitheliomas in patients with the cancer predisposition Bazex–Dupré–Christol Syndrome [101]. In the second mechanism, disruption of TAD structure leads to aberrant contacts between enhancers and gene promoters that are normally insulated by TAD boundaries, allowing distal enhancers to ectopically activate disease-relevant genes. This rearrangement was termed as “enhancer hijacking”, which might be prevalent in a variety of congenital diseases. Although robust estimates of the frequency are unavailable, analysis of causal SNVs in a limb malformation cohort demonstrated that position effects, potentially underlying disease mechanisms, were present in nearly half of cases [102].

### 4.2. Gene Co-Expression

3D genome organization plays a crucial role in modulating gene co-expression—a program that ensures synchronized transcriptional activity of genes within and across genomic regions. Within local genomic regions, TADs serve as key chromatin structural units that facilitates coordinated gene expression. Genes located within the same TAD often display synchronized transcriptional dynamics during differentiation [103], suggesting that TADs help organize neighboring genes into co-regulated units. TAD boundaries are enriched with regulatory and architectural features, including epigenetic marks, transcription start sites, CTCF binding sites and proteins [39,42,104], which establish structural barriers that restrict cross-talks between adjacent TADs while reinforcing intra-domain regulatory interactions. Perturbations of these boundary elements can disrupt TAD structures, which leads to alterations in gene expression patterns [41,105,106]. Beyond local chromatin domains, long-range regulatory interactions are also capable of bringing distant enhancers into proximity with gene promoters, thereby ensuring precise gene expression patterns during cell differentiation and development [107]. These interactions are traditionally thought to occur within TADs and reinforce intra-domain regulatory specificity. However, emerging studies have indicated that enhancers can also establish interactions that cross TAD boundaries, contributing to the co-expression of multiple genes in larger genomic regions [108,109,110].

In addition, transcriptional condensates represent another mechanism of long-range transcriptional coordination. These condensates arise through weak, multivalent interactions among intrinsically disordered protein domains, enabling the spatial clustering of co-regulated genes within specific nuclear regions [111,112,113]. Live-cell imaging has revealed that these structures are highly dynamic, forming de novo upon transcriptional activation and dissolving within minutes [114]. This transient assembly allows genes to dynamically associate and dissociate from condensates, thereby facilitating transcriptional bursting and rapid responsiveness to regulatory signals. By creating specialized microenvironments for the simultaneous activation of multiple genes, these condensates enhance the efficiency and coordination of transcription. For instance, coactivators like BRD4 and Mediator proteins have been shown to form phase-separated condensates at super-enhancers, which compartmentalize and concentrate the transcription apparatus shared by key lineage-specific genes [115,116]. Interestingly, these regulatory effects by phase condensates seem to be counteracted by the architectural proteins, since cohesin depletion promotes spatial mixing of transcriptional hubs across distinct regulatory domains and increases intra-chromosomal gene co-bursting, highlighting a dynamic balance among different regulations of genome structure in controlling gene co-expression [117].

### 4.3. Epigenetic Mark Deposition

Major epigenetic regulatory events, such as histone modifications and DNA methylation, are not confined to the linear genome but are deeply embedded within the 3D architecture of chromatin [118]. This spatial organization provides a physical framework for the establishment, stability, and functional interpretation of these genome-associated epigenetic marks. Recent theoretical studies have proposed that 3D genome folding can determine the spatial range of epigenetic mark propagation and facilitate the formation of functional chromatin domains. Based on a generic polymer 3DSpreader model, Katava et al. showed that histone modifications, such as H3K9me3, can spread across distal genomic regions through spatial chromatin contacts [119]. Unexpectedly, this model not only predicted but also mechanistically explained how domain formation and boundary stability can emerge even in the absence of predefined insulator elements, provided that the propagation of epigenetic modifications proceeds more slowly than chromatin relaxation dynamics. More recently, Owen et al. developed a model based on prior experimental findings, predicting that the three-dimensional folding of the genome enables cells maintain their identities over successive passages by dynamically preserving and redistributing epigenetic marks [8]. This model provides a unified framework that reconciles a wide range of observations, from classic phenomena such as position-effect variegation to more recent insights into the dynamics of epigenetic mark recovery after DNA replication. Although tempting, direct experimental validation of this model remains challenging due to the current lack of tools capable of measuring epigenetic dynamics at the single-cell level.

### 4.4. DNA Replication

DNA replication in eukaryotic cells occurs at specific nuclear structures, where it forms novel ring-like architectures known as “replication clusters” [120].These clusters exhibit distinct spatial distributions within the nucleus depending on the stage of replication. With the advent of high-resolution techniques such as Hi-C and Micro-C, it has been revealed that early DNA replication predominantly occurs in transcriptionally active A compartments, while late replication tends to localize to B compartments at the chromatin compartment level [121]. In early S phase, replication clusters are distributed throughout the nucleoplasm, whereas in late S phase, they are predominantly found near the nuclear periphery and nucleoli—regions enriched in B compartments. Together, these observations underscore a strong correlation between three-dimensional genome architecture and replication timing. At finer resolution, studies in mouse ESCs revealed that weakening or inverting TAD boundaries does not significantly alter replication timing, indicating that TAD boundaries are not direct determinants of this process [122]. However, acute degradation of cohesin leads to dramatic widespread reconfiguration of genome organization, resulting in numerous DNA double-strand breaks and earlier replication initiation in approximately 30% of genomic regions [123]. Notably, a recent study using Repli-HiC—a method for capturing chromatin interactions involving nascent DNA—revealed that replication forks are coupled through vertically striated, fountain-like spatial chromatin structures during DNA synthesis [124]. This finding provides new insight into the spatial dynamics of DNA replication at high resolution.

### 4.5. DNA Repair

In addition to DNA replication, DNA damage repair is also intricately regulated by 3D organization of the genome. It is generally accepted that repair efficiency is closely associated with local chromatin context, since repair proteins can readily access damaged sites in relaxed euchromatin regions, leading to more efficient repair. In contrast, the compact and highly condensed structure of heterochromatin hinders the accessibility of repair machinery and consequently impairs repair efficiency. To overcome this limitation, cells have evolved sophisticated mechanisms to facilitate double-strand break (DSB) repair in heterochromatin. For example, DSBs can be actively relocated from heterochromatic regions to more open chromatin regions to enhance repair efficiency [125]. Moreover, Hi-C and 4C-seq analyses further revealed that DSB formation induces increased insulation at damaged TAD boundaries, which helps confine the damage site and promotes repair [126]. Following induction of DSB in mammalian cells, the formation of new nuclear compartments facilitates the enrichment of DNA damage response factors, thereby upregulating the expression of repair genes and promoting efficient damage response [25]. Recent studies have shown that genome folding plays a crucial role in the response to UV-induced DNA damage. Specifically, after UV irradiation, TAD boundaries are reinforced and exhibit higher nucleotide excision repair activity, along with increased DNA accessibility and regulatory activity [127]. Interestingly, repair activity is significantly elevated at loop anchor regions compared to loop flanks and loop bodies, with repair efficiency correlating with loop strength, DNA accessibility, and regulatory activity [127].

## 5. 3D Genome Architecture in Differentiation, Development and Diseases

3D genome architecture serves as a critical determinant of spatiotemporal-specific gene regulation, fundamentally governing cell differentiation, organismal development, and disease pathogenesis. Recent advances in Hi-C, single-cell multi-omics, and spatial transcriptomics have progressively unveiled the mechanisms by which hierarchical genome structures orchestrate these fundamental biological processes.

### 5.1. Cellular Differentiation

3D genome architecture dynamically reorganizes during cellular differentiation to orchestrate cell type-specific gene expression. Chromatin loops play a central role by modulating E-P interactions, directly regulating lineage-specific transcriptional programs. For example, CTCF-mediated E-P loop structures, together with the erythroid-specific transcription factor GATA1 and CTCF, repress the expression of certain lineage-specific genes, thereby affecting the differentiation of common myeloid progenitor cells into T, B or myeloid cells [128]. In mouse ESCs, loss of KLF4 disrupts the proper formation of E-P loops, resulting in abnormal activation of genes that inhibit neural differentiation, ultimately impairing the neuronal differentiation process [129]. Beyond chromatin loops, the reinforcement of TAD boundaries during early T cell differentiation effectively insulates aberrant E-P interactions and maintains the expression of key lineage-specific transcription factors, thereby safeguarding the T cell differentiation program [130]. Notably, approximately 10–40% of TAD boundaries exhibit cell-type specificity during human ESC differentiation, suggesting that remarkable reorganization of these structures throughout this process [131]. Furthermore, compartment dynamics also contribute to lineage specification, as approximately 36% of the genome undergoes compartment switching between A and B compartments during human ESC differentiation [131], potentially balancing the pluripotency and differentiation capacity of stem cells [44].

### 5.2. Development

The spatiotemporal dynamics of 3D genome architecture precisely regulate stage-specific gene expression programs during various developmental processes. Extensive studies have illustrated the critical interplay between genome structure and early embryonic development, demonstrating how the spatial organization of DNA is dynamically reprogrammed to drive the incredible transformation from a single fertilized egg to a complex multicellular embryo [43,132,133]. Using low-input in situ Hi-C (sisHi-C), researchers discovered that sperm retain higher-order chromatin structures, including TADs and compartments, whereas MII oocyte chromatin lacks such defined structures and exists in a more homogeneous state [43]. After fertilization, the parental genomes undergo rapid decompaction into a relaxed conformation, followed by gradual 3D reorganization during early development, which lays the structural foundation for embryonic gene expression [43]. In human embryos, A/B compartments are absent at the 2-cell stage but progressively reestablish during the 4- to 8-cell transition [132]. Concurrently, TAD boundaries become increasingly defined from initially weak organization at the 2-cell stage [132]. Complementary murine studies further reveal that dispersed chromatin in 1-cell embryos facilitates expression of totipotency-associated genes, and by the 8-cell stage, chromatin reopens into a fibrous network architecture that enables the precise spatiotemporal expression of pluripotency-establishing genes [134]. During blastocyst formation, totipotent cells undergo lineage specification into inner cell mass (ICM) and trophectoderm (TE), a process regulated by pioneer transcription factors such as OCT4 and SOX2 [135]. Interestingly, optimized CUT&RUN experiments demonstrated that SOX2 orchestrates 3D chromatin dynamics through multifaceted SOX2-chromatin interaction modes—“Settler”, “Pioneer”, and “Pilot”—rather than acting solely as a simple pioneer factor. Such multifaceted chromatin interacting modes underscore the critical role of genome structure in response to TF regulation in maintaining pluripotency [135].

Another notable developmental paradigm involving 3D genome reorganization is X chromosome inactivation (XCI) in female mammals. To achieve dosage compensation through XCI, one X chromosome undergoes silencing, accompanied by dramatic remodeling of compartments, TADs, and chromatin loops [136,137]. sisHi-C studies revealed that imprinted XCI initiates from 4-cell to blastocyst embryos, followed by transient reactivation and subsequent random XCI [138]. During inactivation process, TAD and compartments weaken, while cohesin recruitment to the Xist Regulatory Region (XRR) drives X-megadomain formation. This dramatic spatial rearrangement facilitates selective gene activation within a globally silenced chromatin environment, ensuring precise dosage compensation and lineage specification [138]. Conversely, failure of XCI can lead to dynamic disruptions in 3D genomic architecture and dose-dependent overexpression of X-linked genes, impairing germ cell differentiation programs and resulting in sex development disorders such as Klinefelter Syndrome (KS) [139].

Beyond early development, 3D genome organization also directs cell fate determination during organogenesis. In lymphopoiesis, IKAROS mediates megabase-scale, cohesin-dependent looping between lymphoid enhancers, overcoming CTCF-mediated insulation to activate lineage-specific genes [140]. During neuronal differentiation, 83% of differential E-P contacts are associated with gene expression changes, implicating the critical role of chromatin looping in developmental gene regulation [141]. TADs also exhibit stage-specific reorganization throughout tissue development. For example, early neuronal development is characterized by fewer, larger TADs that promote gene expression, whereas later stages show more defined TAD boundaries that insulate transcriptional domains and restrict developmental gene expression [141]. Conversely, disruption of TAD boundaries leads to pathogenic enhancer hijacking–as exemplified by the *EPHA4* locus, where boundary deletion creates ectopic limb enhancer contacts, resulting in congenital limb malformations [105]. Additionally, early neuronal differentiation is marked by widespread compartment switching from B to A, repositioning neuronal genes into transcriptionally active compartments. This reorganization suggests that compartment transitions play a key role in refining the transcriptional landscapes to support lineage-specific gene expression [141].

### 5.3. Diseases

Aberrations in 3D genome architecture are closely associated with a variety of pathological processes, including developmental defects, neurodegenerative diseases, and malignant transformation (Figure 3). These structural anomalies frequently disrupt gene expression programs by altering chromatin looping, TAD boundaries, and compartmentalization.

In developmental disorders, recent findings have revealed that chromatin loops regulate the expression of XCR1 (X-C motif chemokine receptor 1), a key G protein-coupled receptor, through long-range chromatin interactions involving the transcription factor RUNX2. This regulation influences osteogenic differentiation and subsequently increases susceptibility to osteoporosis [142]. During murine T cell differentiation, defects in polyamine metabolism induce excessive condensation of B compartments and decompaction of A compartments, disrupting the gene regulatory networks essential for T cell development [38]. In neuro-developmental disorders, aberrant compartmentalization of the autism candidate gene *CADPS2*, along with heterochromatin disorganization mediated by neuron-specific H4K20me3 modifications, directly triggers transcriptional dysregulation and ultimately contributes to disease onset [143].

In neurodegenerative diseases, disruption of chromatin loop structures within the protocadherin (*Pcdh*) gene cluster in neurons results in dysregulation of neuron-specific gene expression and impaired synaptic plasticity, ultimately contributing to the onset of autism spectrum disorder (ASD) [144]. In Fragile X syndrome, which is caused by CGG repeat expansion, patients exhibit severe TAD folding abnormalities in neurons, leading to neuronal dysfunction and disease manifestation [145]. In microglia-mediated models of neurodegeneration, integrated analyses using ATAC-seq and promoter-capture Hi-C (pcHi-C) revealed significant remodeling of the E-P interaction network in infected microglia [146]. This remodeling potentially facilitates the formation of aberrant chromatin loops between distal candidate cis-regulatory elements and promoters of genes such as *MS4A6A* [146]. These 3D structural anomalies eventually drive the upregulation of target genes, aligning with the transcriptional signature of disease-associated microglia, thereby confirming that abnormal E-P looping contributes to the risk of neurodegenerative disorders [146].

In malignancies, increasing evidence links disruption of 3D genome architecture to tumor initiation and progression. For instance, in metastatic triple-negative breast cancer (TNBC), a tandem dual-loop structure—connecting enhancers, promoters, and gene termination regions—is mediated by the enhancer activity regulatory protein CREPT, drives distal metastasis of TNBC by amplifying metastasis-associated genes expression [147]. Similarly, in metastatic pancreatic cancer, the number of metastasis-specific E-P loops is significantly increased, leading to the upregulation of oncogenic target genes. This aberrant transcriptional activation is strongly related to poor prognosis, conferring TNM-independent prognostic risk [148]. Apart from aberrant chromatin looping, TAD boundary disruption and compartment switching also contribute to carcinogenesis. A systematic analysis of 8928 samples across 33 cancer types revealed that TAD misfolding significantly increases the risk of various cancers, including solid tumors such as glioma and breast cancer, as well as hematological malignancies [149]. Multiple myeloma (MM), a malignant hematologic tumor derived from plasma cells, is characterized by recurrent genomic aberrations, including frequent aneuploidy, copy number variations (CNVs), and chromosomal translocations [150]. Integrative analyses of Hi-C, whole-genome sequencing, and transcriptome data have revealed extensive reorganization of the 3D genome architecture in MM, marked by an increased number of TADs and a reduction in their average size [150]. Furthermore, A/B compartment states have been proposed as a reliable predictor of genetic subtypes in acute myeloid leukemia (AML). Genomic analyses of primary AML patient samples have revealed abnormal A/B compartment switching, which may induce the dysregulation of disease-related genes such as *WT1*, ultimately leading to aberrant myeloid differentiation [151]. In addition, tumors with loss-of-function mutations in the SWI/SNF complex often exhibit abnormal compartment stabilization (rigidification). This aberrant chromatin rigidity represses the expression of tumor suppressor genes like *p16*, thereby facilitating to the development of both neurodevelopmental disorders and malignancies, while potentially conferring therapy resistance [152].

In summary, 3D genome organization directly underlies the onset and progression of various diseases. Deciphering the multi-scale dynamics of chromatin architecture will provide novel insights into diverse disease etiologies and facilitate the development of more precise and effective therapeutic strategies.

## 6. Conclusions and Future Perspectives

The 3D genome architecture is not merely a passive physical attribute, but is intimately linked to essential biological processes in mammals. Through the formation of chromatin compartments, TADs, and finer-scale chromatin loops, the genome maintains precise modulation of transcriptional activity and gene expression. Importantly, this 3D organization is highly dynamic and can be rapidly restructured in response to developmental cues, environmental stimuli, and changes in cellular state, allowing for timely spatial reconfiguration and adaptive gene regulation. In recent years, a growing body of evidence has linked aberrant 3D genome organization to various human diseases, highlighting its critical biological and pathological relevance. Although research over the past fifteen years has shed light on the fundamental principles, regulatory mechanisms, and functional roles of genome folding, this rapidly evolving field, as is often the case in frontier science, has generated more questions than definitive answers.

Firstly, although an increasing number of regulatory factors have been identified to orchestrate genome organization, the biophysical processes through which they coordinate and cooperate remain elusive. Future studies on genome folding mechanisms will increasingly focus on decoding the biophysical principles underpinning nuclear organization, particularly the role of biomolecular phase separation in forming functional compartments such as nuclear speckles, super-enhancer clusters and heterochromatin compartments [68,69,153,154]. In parallel, evolutionary comparative studies will be employed to explore the conservation of these organizational principles across species [38].

Secondly, while considerable progress has been made in understanding the mechanisms underlying genome folding, its functional consequences remain poorly understood. Future efforts to investigate the biological function of genome organization will shift from descriptive and correlative mapping toward establishing direct causality and mechanistic understanding. This paradigm shift will be propelled by next-generation perturbation tools that enable precise manipulation of specific architectural features—such as disrupting individual chromatin loops or dissolving TAD boundaries via CRISPR-Cas9 genome editing tools [155], depleting architectural regulators via targeted degron systems [156], or opto-genetically inducing artificial compartments [157]—coupled with real-time monitoring of functional outcomes in living cells [158,159]. Meanwhile, translational research will increasingly focus on how non-coding variants and structural mutations distort 3D structure to drive disease pathogenesis [160], leveraging patient-derived organoids and isogenic cell models as physiologically relevant platforms [161,162,163]. 

Lastly, in order to gain a comprehensive understanding of both the biophysical nature of genome organization and its biological consequences, integrative multi-omics approaches will become indispensable, combining ultra-high-resolution 3D genomics (e.g., Micro-C [11], in situ Hi-C [164]), single-cell epigenomics, single-cell transcriptomics, and complemented with super-resolution imaging. To manage the resulting data complexity, artificial intelligence (AI)-driven computational modeling will be essential for predicting the functional impacts of structural variants and integrating massive multidimensional datasets [165,166]. This convergence will enable a comprehensive understanding of how spatial genome architecture regulates not only gene expression but also DNA repair fidelity, replication timing, and lineage-specific differentiation trajectories across diverse cellular states and environmental responses. Ultimately, these synergistic advances promise to transform our perception of the nucleus—from a passive container to a dynamically orchestrated, functionally integrated processor—where genome architecture acts as a central regulator of cellular homeostasis, organismal development and disease pathogenesis.

## Figures and Tables

**Figure 1 ijms-26-09058-f001:**
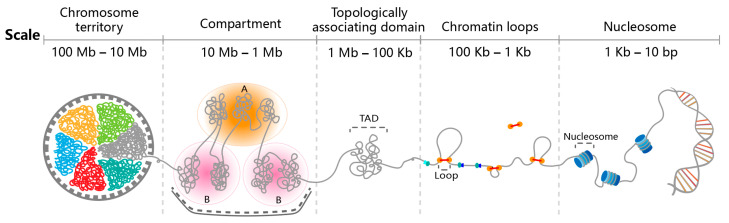
Hierarchical organization of the 3D genome. Within a single nucleus with ~20 μm diameter, the mammalian genome is folded into highly ordered structures: The basic genomic unit is the nucleosome—every 146 bp DNA wrap around a histone octamer—resembling beads on a string; then necklace-shaped genomic DNA continuously extrude chromatin loops driven by ring-shaped cohesin complexes; an array of chromatin loops give rise to topologically associating domains (TADs), which are genomic regions exhibiting more frequent intra-domain than inter-domain chromatin interactions; above TADs, the genome segregates into transcriptionally active A compartments and transcriptionally repressive B compartments; finally at the chromosomal level, each chromosome occupies a relatively separated nuclear space, named “chromosome territory”.

**Figure 2 ijms-26-09058-f002:**
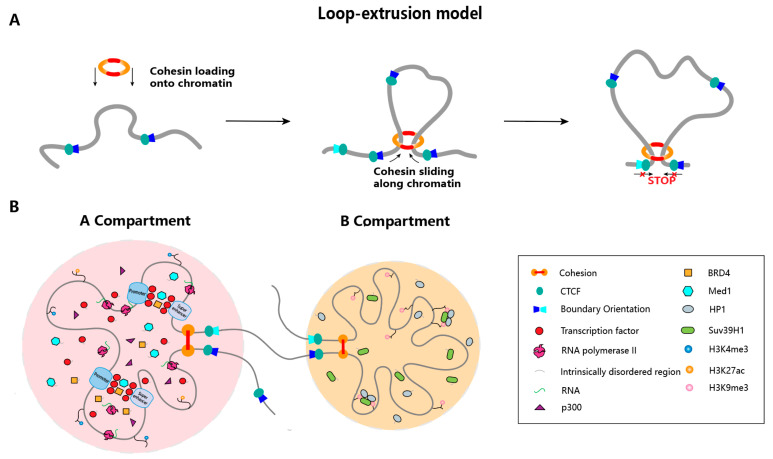
Loop extrusion and phase separation-driven organization of the 3D genome. (**A**) According to the loop extrusion model, cohesin complex progressively extrudes chromatin to form loops. This extrusion process is anchored and paused by CTCF bound to convergent binding sites. CTCF sites occur genome-wide in convergent, divergent, or same-direction orientations, contributing to different consequences in controlling extrusion progression. (**B**) Schematic indicates phase separation-driven organization of A and B compartments. In A compartment, factors such as TFs, Mediators, epigenetic regulators and enhancer RNAs coalesce via phase separation to assemble transcriptional condensates, thereby enabling the bridging of super-enhancers and promoters for robust activation of key developmental genes. In B compartment, HP1 dimers recruit histone methyltransferases such as SUV39H1 to augment H3K9 trimethylation and bridge adjacent nucleosomes bearing H3K9me3, driving the formation of heterochromatin phase condensates that effectuates pronounced chromatin compaction.

**Figure 3 ijms-26-09058-f003:**
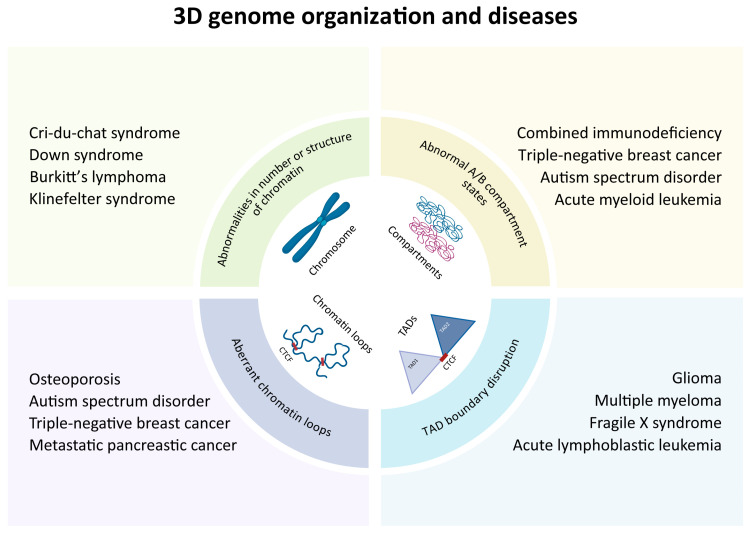
Overview of multiscale 3D genome architectural aberrations and associated diseases. Aberrations in genome architecture across multiple hierarchical levels contribute to a wide range of pathological conditions, including developmental abnormalities, neurodegenerative diseases, and cancer. Multiscale spatial aberrations and their associated diseases: abnormalities in number or global structure of chromatin (e.g., Cri-du-chat syndrome, Down syndrome, Burkitt’s lymphoma and Klinefelter syndrome); abnormal A/B compartment states (e.g., combined immunodeficiency, triple-negative breast cancer, autism spectrum disorder and acute myeloid leukemia); aberrant chromatin loops (e.g., osteoporosism, autism spectrum disorder, triple-negative breast cancer and metastatic pancreatic cancer); TAD boundary disruption (e.g., glioma, multiple myeloma, fragile X syndrome and acute lymphoblastic leukemia).

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
