# Peer review of "The Biological Function of Genome Organization"

_ijms, 2025, doi:10.3390/ijms26189058_

Round 1
Reviewer 1 Report
Comments and Suggestions for Authors
The article “The biological function of genome organization” is a review that discusses the hierarchical 3D organization of the mammalian genome, exploring how its structure regulates key nuclear functions, influences biological processes like transcription and DNA repair, changes during development, and impacts human diseases.
Although this manuscript could be of interest to readers, it contains several major flaws that need to be addressed before publication. I was initially unsure whether to reject it or request major revisions, but after careful consideration, I have decided to recommend major revisions. The authors are expected to address the following concerns:
- The English style should be improved; I recommend a revision of the style.
- It is not clear the aim of this review. It looks like a book chapter rather than a review article.
- The authors should clearly highlight the novel elements and distinct contributions of this review compared to the existing reviews available in literature.
- Considering the amount of literature, the review is excessively short; authors should include more information about this topic.
- I was surprised finding that authors completely ignored a master mechanism that regulates chromatin remodeling and condensation. SIRT1, a NAD⁺-dependent deacetylase, influences chromatin condensation by removing acetyl groups from histone proteins, particularly histone H3 and H4. This deacetylation reduces the negative charge on histones, promoting tighter interaction between histones and DNA, which leads to a more compact and condensed chromatin structure. By enhancing chromatin condensation, SIRT1 plays a key role in regulating gene expression, DNA repair, and maintaining genomic stability (PMID: 39273039). Moreover, NAD⁺ homeostasis is closely linked to chromatin remodeling because NAD⁺ serves as an essential cofactor for several chromatin-modifying enzymes, particularly sirtuins (like SIRT1) and poly(ADP-ribose) polymerases (PARPs). These enzymes use NAD⁺ to catalyze reactions that modify histones or chromatin-associated proteins, thereby altering chromatin structure and accessibility. (PMID: 36829935). Thus, the master regulator of nicotinamide intracellular content (nicotinamide N-methyltransferase, NNMT) may also affect indirectly the chromatin remodeling by influencing NAD-dependent enzymes, since nicotinamide can be converted into NAD by the NAD-salvage pathway (PMID: 34153425).
- A paragraph summarizing synthetic and natural compounds able to impact on chromatin condensation/remodeling and epigenetic is mandatory.
- The conclusion section fails to provide information about challenges and future directions based on what reported in the manuscript.
Reviewer 2 Report
Comments and Suggestions for Authors
This kind of comprehensive review article creates current knowledge on the hierarchical organization of the mammalian genome and its critical biological functions. The authors effectively outline the multi-scale architecture, from chromatin loops and topologically associating domains to compartments and chromosome territories, and discuss the key regulators, including architectural proteins, transcription factors, epigenetic modifiers, and non-coding RNAs. The review delves into the functional consequences of this 3D organization, detailing its role in transcription, gene co-expression, DNA replication, and repair. It further explores the dynamic nature of genome architecture during development and differentiation and its disruption in various human diseases. The article concludes by highlighting future directions in the field.
I consider the review is timely, well-structured, and covers a vast and complex field with considerable clarity. it will be very useful for especially for new scientists and postgraduate students as well postdoctorals.
However, I have some suggestiions and notes for authors
- The heavy use of acronyms can be challenging to follow, especially for non-specialists or students. I suggest to have a supplementary table listing all major acronyms.
_ For me, i found some sections are dense, for example 3.2 and 4.2, the descriptions of protein interactions and condensate dynamics could be slightly streamlined for clarity.
Round 2
Reviewer 1 Report
Comments and Suggestions for Authors
The authors did not address properly any of the raised concerns. All that was made is a cosmetic refresh of the paper.
English revision was recommended and was not performed.
The aim of the review and the novel distinct contribution was not deeply discussed since the explaination that authors gave does not justify this work which does not add anything to the existing literature and is just another copy of reviews on the same subject.
The genome organization topic requires a deeper discussion, not only few pages including figures, it is partial and incomplete.
The paragraph regarding natural and synthetic compounds on chromatine and epigenetic remodeling was not added.
The conclusion section still fails to provide perspectives.
None of the point was addressed, the manuscript is rated very low quality and must be rejected.
Author Response
The editor has made a decision.